# miRNAs in Cancer (Review of Literature)

**DOI:** 10.3390/ijms23052805

**Published:** 2022-03-03

**Authors:** Beata Smolarz, Adam Durczyński, Hanna Romanowicz, Krzysztof Szyłło, Piotr Hogendorf

**Affiliations:** 1Laboratory of Cancer Genetics, Department of Pathology, Polish Mother’s Memorial Hospital Research Institute, Rzgowska 281/289, 93-338 Lodz, Poland; hanna-romanowicz@wp.pl; 2Department of General and Transplant Surgery, N. Barlicki Memorial Clinical Hospital, Medical University of Lodz, 90-001 Lodz, Poland; adam.durczynski@umed.lodz.pl (A.D.); piotr.hogendorf@umed.lodz.pl (P.H.); 3Department of Gynaecology, Oncological Gynaecology and Endometriosis Treatment, Polish Mother’s Memorial Hospital Research Institute, Rzgowska-Lodz, 93-338 Lodz, Poland; kszyllo@o2.pl

**Keywords:** cancer, miRNA, diagnosis

## Abstract

MicroRNAs (miRNAs) are short, noncoding, single-stranded RNA molecules that regulate gene expression at the post-transcriptional level by binding to mRNAs. miRNAs affect the course of processes of fundamental importance for the proper functioning of the organism. These processes include cell division, proliferation, differentiation, cell apoptosis and the formation of blood vessels. Altered expression of individual miRNAs has been shown in numerous cancers, which may indicate the oncogenic or suppressor potential of the molecules in question. This paper discusses the current knowledge about the possibility of using miRNA as a diagnostic marker and a potential target in modern anticancer therapies.

## 1. Introduction

Cancer is an important problem in this day and age. Scientists are constantly searching for new factors responsible for the process of carcinogenesis. MicroRNAs (miRNAs) are a group of small single-stranded RNA molecules involved in regulating the expression of many genes preserved in evolution. In humans, an miRNA molecule is most often 22 nucleotides long, although molecules with a length of 19–25 nucleotides have been described [1].

miRNAs control biological processes, such as cell division, cell differentiation, angiogenesis, migration, apoptosis and oncogenesis [2,3,4,5]. Genes encoding miRNAs are located in the introns or exons of protein-coding genes and in intergenic regions, often in regions with high instability. They are characterized as noncoding RNAs.

## 2. miRNA

In 1993, Victor Ambros and Gary Ruvkun, while working on *Caenorhabditis elegans*, made a revolutionary discovery that ushered in a new era. They observed a relationship between the amount of LIN14 protein and 22-nucleotide RNA, encoded by the LIN-4 gene, which is involved in the development of *C. elegans*. The discovery of this post-transcriptional silencing of target mRNA by small RNA molecules changed the understanding of the system controlling the expression of genetic information [6,7,8,9]. A dozen or so years later, in 2005, the participation of miRNA molecules in the cancer process and the possibility of determining specific miRNA profiles, which in the future could be used as diagnostic markers, were suggested [10,11]. In a groundbreaking study on a mouse xenogenous model of prostate cancer, the relationship between the amount of miRNA in the blood and the size of the cancer that was transplanted was revealed [12]. Other repeatable studies have confirmed the thesis that extracellular miRNAs are detectable in blood serum and that their concentration is at a similar level in healthy people. Changes in the level of miRNA molecules may be a consequence of various physiological conditions, such as pregnancy (placental miRNAs are present in the blood serum of pregnant women, which can be used to determine the stage of pregnancy) or various types of diseases and viral infections, as well as cancer diseases. It has been suggested that a specific miRNA pattern (increase or decrease in expression relative to control), both in blood and tissue, may be characteristic of a given disease, and this allows the use of selected miRNAs to monitor the physiological state of patients. The use of miRNA in diagnostics has significantly expanded due to the discovery of their presence in other body fluids, such as urine, blood, bronchial lavage, synovial fluid, milk, saliva and cerebrospinal fluid [13,14]. A single miRNA can modulate thousands of genes by recognizing complementary sequences at the 3’ UTR end of the target mRNA. It is estimated that about 30% of human mRNAs are strictly regulated by miRNAs, but this figure increases after taking into account reports indicating the possibility of some miRNAs also binding to the 5’ UTR region and the open reading frame (ORF) region (however, in such cases they occur less frequently and work less effectively) [15]. Endogenous miRNAs affect processes in cells such as proliferation, DNA repair, cell differentiation, metabolism and apoptosis; however, the biological function of circulating miRNAs needs to be investigated. It is assumed that some extracellular miRNAs may be carriers of information between cells during many physiological and pathological processes [16]. Depending on which genes they influence, miRNAs can function as oncomirs—procancerogenic or suppressor miRNAs—that inhibit oncogenes [17]. Mechanisms, e.g., genetic changes (point mutations and single-nucleotide polymorphism of SNPs) and epigenetic changes within miRNA genes can affect their expression and thus lead to changes in the expression of target genes. Genes encoding miRNAs are often located within fragile chromosome sites, areas with a lack of LOH heterozygosity and minimal amplification regions, supporting the theory of miRNA’s relationship with the tumor process. Quantitative changes in mature miRNA molecules may also be caused by abnormalities associated with proteins involved in the biogenesis of miRNA molecules [18].

Previous research results indicate a link between regulatory disorders in the expression of relevant miRNAs and the occurrence of various types of cancer [19]. Thanks to the development of research on microRNAs, the possibility of typing cancers has appeared. The procedure is based on the identification of miRNAs specifically expressed in tumor tissues or miRNAs whose expression level is different from that present in normal tissues. The profile of differentiation and expression of miRNA allows the degree of tumor development to be determined, which clarifies the therapeutic possibilities and would allow the use of the most appropriate therapy for a given case. The discovery that miRNAs are markers of the tumor process and do not require invasive diagnostic procedures is extremely promising for diagnostics. Intensive research is underway on the use of miRNAs present in body fluids (such as plasma, cerebrospinal fluid, saliva, urine, seminal fluid) as a diagnostic marker or prognostic marker of cancer (Table 1).

## 3. miRNAs and Their Role in Oncogenesis

miRNAs regulate the expression of about 60% of human genes [90]. Interestingly, one miRNA molecule (Up to 2019, about 2300 different miRNAs have been described in humans [91]) can attach to many target mRNAs. In turn, one mRNA molecule can be inhibited by different miRNAs. The effect of interaction with target mRNA molecules depends on the complementarity of the bond and the level of expression of miRNA or mRNA [91]. miRNA expression disorders consisting in the lack of expression of a specific miRNA or expression of miRNA not yet present in this tissue or an increase or decrease in the expression of selected miRNAs have been shown to occur in the course of many diseases, including cancer [85].

miRNA expression disorders in cancer cells are often rooted in the localization of the genes encoding them. They are often located in genetically unstable regions, fragile sites or cancer-associated genomic regions (CAGRs), which often results in their deletion, resulting in a lack of miRNA expression [92].

For many years, it was thought that the expression of miRNA in cancer cells was primarily reduced. Only a comparison of the miRNA profile of normal and cancerous tissues showed significant overexpression of some miRNAs [93]. Depending on the function of miRNAs in the development of tumors, they are classified as: suppressor miRNAs (inhibiting the expression of oncogenes or genes that induce apoptosis) and oncogenic miRNAs (activating oncogenesis or inhibiting the expression of suppressor genes) [94]. It should be emphasized that this classification is a significant simplification, because in the case of many miRNAs (e.g., miR-155, miR-125b), the effect of their activity depends on the total activity of regulated genes [94].

Decreased expression or lack of expression of suppressor miRNAs results in increased expression of genes important for tumor progression, including antiapoptotic proteins or transcription factors. In 2017, reduced expression of miR-15 and miR-16 molecules in chronic lymphocytic leukemia (CCL) cells was first described, leading to inhibition of the apoptosis process (miR-15 and miR-16 regulate the expression of antiapoptotic BCL-2) and thus contributing to uncontrolled proliferation of leukemic cells [94]. The reduced expression of the miR-146a molecule regulating the expression of the transcription factor NFκB in stomach cancer cells correlates with tumor growth [94]. An in vitro model of colorectal cancer showed a decrease in the expression of miR-143, whose target gene is m.in. Oncogene Raf1 [95]. 

An important role in the regulation of the expression of suppressor miRNAs in cancer cells is played by the process of DNA methylation [96], e.g., hypermethylation of miRNA promoters (let-7, miR-34, miR-342, miR345, miR-9, miR-129, miR-137) leads to a reduction in their expression and the development of colorectal cancer. Decreased expression of miR-143 in colorectal cancer cells results in increased activity of methyltransferase 3A DNA (DNMT3A) and increased proliferation of cancer cells [97]. It is worth emphasizing that miRNA expression, on the one hand, is dependent on DNA methylation, and on the other, it can affect the activity of epigenetic regulators such as DNA methyltransferases or histone deacetylases. Examples of suppressor miRNAs with potential target genes are shown in Figure 1. 

In colorectal cancer cells, increased expression of selected miRNAs is more often observed, which means that they are more often oncogenic in nature [98]. Increased miRNA expression may result from the amplification of genes encoding miRNAs as well as more efficient biogenesis, constitutive activity of their promoters or greater stability of miRNA molecules [96]. Over the last decade, a number of miRNA molecules involved in the initiation, progression and metastasis of breast cancer have been identified [99]. The relationship between the expression of individual miRNAs and the clinical–pathological features of breast cancer or the response to causal treatment of this malignant tumor has also been confirmed [99]. For example, studies have shown that in triple-negative breast cancer there is an overexpression of oncogenic molecules miR-21, miR-210 and miR-221, which is associated with a shorter disease-free time and worse survival [100]. Molecules with reduced expression, and therefore reduced suppressor potential, included, for example, miR125-b in the case of HER-2-positive cancers or miR-520 in hormone-dependent cancers [100]. Singh and Mo presented miRNA families in their review article, which play an important role in the course of the discussed malignant tumor. They focused on the miR-10 family, in which miR-10a and miR-10b are involved in the development and metastasis of breast cancer [100]. MiR-10b overexpression is associated with a higher degree of TNM cancer (larger size of the primary tumor, presence of metastases in the lymph nodes), a greater degree of cellular proliferation and overexpression or amplification of the HER-2 receptor [100]. However, it is negatively correlated with the presence of steroid receptors and the concentration of E-cadherin, which seems to play a role in suppressing the metastasis process in the EMT mechanism. (epithelial–mesenchymal transition) [100]. Metastasis, as well as a worse course of particularly ductal breast cancer and consequently shorter overall survival, is also associated with the oncogenic miR-21 family [100]. Among the families of suppressor miRNAs with reduced expression in cancerous breast tissue compared to healthy tissue, the aforementioned authors mentioned the miR-200 family and miR-205 and miR-145. miR-200 and miR-205 probably inhibit the metastasis process associated with the EMT mechanism, and miR-145 affects cell apoptosis [100]. On the other hand, in a 2019 review article by Loh et al., the decisive oncogenic potential of the miR-200 family was described. Increased concentrations of individual miR-200s were associated not only with breast cancer’s ability to form distant metastases, but also with resistance to chemotherapy [99]. Increased expression of miRNA leads to the repression of numerous genes with a suppressor effect. An example is “oncomiR-1” or a cluster of six miRNAs (miR-17-92; miR-17, miR-18, miR-19a, miR-20, miR-19b and miR-92), which inhibits the expression of the Rbl2 suppressor gene [93]. Oncogenic miRNAs, e.g., miR-24, miR-31 and miR-21, increase the proliferative potential of cells by silencing CDK inhibitors (cyclin dependent kinases) [93,100]. Table 2 shows examples of oncogenic miRNAs and the genes they regulate [101,102,103,104]. 

## 4. Circulating miRNA

The first circulating miRNAs were detected in the circulatory system of people with diffuse large B-cell lymphoma. miR-21, miR-10 and miR-155 can be reliably determined in blood serum and allow for the differentiation between sick and healthy people, and because they show high stability, they can be used in clinical diagnostics [105]. Along with the development of research, the growing potential of miRNA molecules in determining the degree of cancer malignancy or predicting the effects of specific therapies, as well as in the treatment of cancer patients, began to be noticed. miRNAs—unlike mRNA—in plasma, serum, fresh-frozen tissues, paraffin blocks and saliva are characterized by resistance to endo- and exogenous RNases, extreme temperatures and pH. miRNAs are characterized by the ability to maintain high stability for a long period of time, even when left at room temperature. From the point of view of clinical diagnostics, these characteristics make miRNA molecules excellent biomarkers, especially when attention is paid to the need to repeatedly freeze and defrost the diagnostic material, which quite often happens when processing laboratory samples [106,107]. Since synthetic, blood-derived and purified miRNAs, when reinjected into human plasma, were susceptible to endogenous RNaz and immediately degraded, the causes of their specific stability have been investigated [108]. So far, there have been many hypotheses explaining the origin and properties of circulating miRNAs, which are closely related to each other and most likely do not exclude each other. Three leading theories are listed. The first is based on the claim that the occurrence of miRNAs in the blood is an undesirable effect of cell destruction and occurs as a result of nonenergy leakage of cellular miRNAs. This can occur as a result of tissue damage characteristic of individual stages of carcinogenesis, cell entry into inflammation, apoptosis or during the formation of metastases. The second hypothesis assumes an active process of releasing miRNA from the cell in a way that depends on microvesicles (MVs). The third theory, on the other hand, talks about the active and selective secretion of miRNA in an independent and MV-free form, which is a consequence of the cell’s response to various stimuli [109]. Microbubbles are of cellular origin and can be released from the cell by fusion with the cell membrane. These include MPs (microparticles) (>100 nm in diameter) and exosomes (50–90 nm in diameter), as well as larger apoptotic bodies (ABs), produced in response to apoptotic stimuli. MVs have been detected in plasma, urine and other physiological fluids, and the ability of MVs to release miRNA allows most cells to function under pathological and physiological conditions [109]. 

Valadi et al. [110], in their pioneering work, were the first to report on the transport of miRNAs via exosomes, which, functioning as a vehicle for transporting miRNAs between cells, can affect their activity and protein production. One strategy for studying circulating miRNAs is to isolate miRNAs associated with exosomes. On such study involving people with non-small-cell lung cancer (NSCLC) was conducted by Rabinowits et al. [111]. There was an elevated level of selected exosome miRNAs compared to the control group. Reduced levels of miR-let-7f, miR-30e-3p and miR-20b associated with MVs compared to levels in healthy people were noted. The high stability of miRNA molecules, in addition to the factors mentioned above, may also correspond to RNA-binding proteins (RBPs) and lipoproteins [112]. The function of miRNA-binding proteins is not only to protect the miRNA molecule from degradation, but also to actively participate in the controlled packaging of specific miRNAs into exosomes and their export outside the cell. The main RBPs include the protein AGO2 (Argonaute 2) belonging to the RISC complex (RNA-induced silencing complex) and NPM1 (Nucleophosmin 1). The miRNA-AGO2 complex is characterized by high stability. However, more research is required to determine the magnitude of the scale at which it can be released from the cell. It is estimated that this fraction may consist of up to 90% of all circulating miRNAs [113,114]. It has been suggested that on the basis of the analysis of specific miRNAs present in the blood and the form they have taken, it is possible to determine which type of cell they come from and thus identify cellularly specific miRNAs or specific methods of their release by a given cell type [115]. 

Examination of miRNAs for specific fractions can significantly increase both the sensitivity and specificity of selected circulating miRNAs as noninvasive biomarkers [20,21,22,23,24,25,26,27,28,29,30,31,32,33,34,35,36,37,38,39,40,41,42,43,44,45,46,47,48,49,50,51,52,53,54,55,56,57,58,59,60,61,62,63,64,65,66,67,68,69,70,71].

Figure 2 shows circulating miRNA as a potential biomarker in various types of cancer.

## 5. microRNA in Cancer Diagnosis and Therapy

miRNAs are seen as potential markers of cancer. First, miRNA molecules are readily available for study because they are present in various body fluids. Secondly, the high biological stability of miRNAs facilitates their detection. Third, miRNAs regulate all stages of tumor development, and in many cases show tissue-specific expression. The use of miRNAs as prognostic and predictive biomarkers during treatment seems to be particularly clinically relevant. For example, elevated expression of miR-21 (circulating in the blood) has a strong association with a predisposition to develop colorectal cancer [116], lung cancer [117,118], breast cancer [119] or pancreatic cancer [120]. miRNA expression in chemoresistant cancer cells may differ from that in cells that are sensitive to chemotherapy [121]; e.g., in colorectal cancer, an increase in miR-21 expression correlates with resistance to fluorouracil therapy due to lowered expression of the repair protein MSH2 [122]. In vitro studies have shown that increased expression of miR-140, miR-215, miR-224 and miR20a promotes the development of chemoresistance to fluorouracil, methotrexate, oxaliplatin or tenipozide in colorectal cancer cells [123,124,125,126]. Selected plasma/serum circulating miRNAs could be used to discriminate various cancer patients from healthy individuals, such as those with breast [127], colorectal [128], gastric [129], lung [130], pancreatic [131] and hepatocellular [132] cancers, making them tools for earlier diagnosis. To date, two tests have been developed to support the diagnosis of cancer based on the miRNA profile. One of them is the ThyraMIR test, which, by assessing the expression of 10 different miRNAs (miR-223-3p, miR-146b-5p, miR-146b, miR-375, miR-31-5p, miR-551b, miR-155-5p, miR-204-5p, miR-138-1-3p, miR-29b-1-5p) allows for the determination of the type of thyroid cancer [133]. 

Another example is the predictive diagnostic test (IVD-certified), miRpredX-31-3p, used in patients with colorectal cancer without mutations in the K-RAS gene. Evaluation of miR-31-3p expression is performed in histopathological scraps from colon tumors. Low expression of miR-31-3p predicts greater clinical efficacy of anti-EGFR therapy vs. traditional chemotherapy [134]. 

Monitoring the changes in the expression profiles of chosen miRNAs could help in early identification of cancer cells and serve as a prediction factor of the disease or treatment. Two therapeutic strategies using miRNAs have been developed that potentially inhibit cancer development (Figure 3).

The first is based on the use of so-called “replacement therapies”, while the second is related to the inhibition of oncogenic miRNAs [135,136]. “Replacement therapies” consist in restoring the expression of suppressor miRNAs, the expression of which is inhibited in cancer cells [137,138]. In this case, artificially synthesized suppressor miRNA molecules should lead to inhibition of the expression of genes that promote the development of tumors (oncogenes) [139]. The strategy of importing exogenous miRNAs may play a role in the treatment of cancer by inhibiting the proliferation or induction of apoptosis of cancer cells [140,141]. An example of replacement therapy is the restoration of miR-34a expression in various types of cancer (e.g., lung, colon, pancreas), which led to inhibition of tumor development and inducing programmed cell death by regulating Notch 1, HMGA2 or Bcl-2 [142]. It is worth mentioning the example of the use of the synthetic miRNA molecule let-7a in the treatment of laryngeal cancer. The introduction of let-7a into cancer cells restored proper regulation of RAS and c-MYC expression and limited the proliferation of cancer cells [143]. A similar result from the use of artificial miRNAs from the let-7 family was observed in the case of hepatocellular carcinoma, where the proliferation and migration of cancer cells was inhibited [144]. Additionally, in the case of colorectal cancer, the miRNA synthetic let-7 contributed to an increase in apoptosis of cancer cells [145]. Simultaneous overexpression of the introduced let-7 and miR-34a molecules inhibited the progression of lung cancer [146]. Furthermore, the introduction of miR-29b molecules into breast or stomach cancer cells reduced tumor growth in vitro and in vivo (in a mouse model), by influencing the Akt3 and KDM2A pathways [147,148]. The introduction of miR-34a into breast cancer cells restores the normal expression of the p53 suppressor protein by regulating the function of the transcription factor Fra-1 [149]. Another example is the molecules miR-15a and miR-16-1, whose dysregulation allows prostate and pancreatic cancers to develop by influencing the signaling pathways associated with CCND1 (cyclin D1), WNT3A and BCL2. 

WNT3A belongs to the Wnt/beta-catenin pathway, which is responsible for cell adhesion and promoting the expression of oncogenes such as c-Myc and CCND1. BCL-2 is responsible for inhibiting apoptosis. Re-expression, i.e., the restoration of normal miR-15a expression, led to a reduction in the viability of cancer cells in in vitro studies [150].

Inhibition of oncogenic miRNAs can be obtained by the introduction of synthetic DNA or RNA molecules that mimic oncosuppressive effects in cancer cells. 

Various variants of inhibitory molecules are used, such as:⁃Anti-miRNA oligonucleotides; AMOs [151], ⁃Locked-nucleic-acid antisense oligonucleotides; LNA; [152], ⁃miRNA sponges; [153],⁃miRNA masks, ⁃Antagomirs; amiRNA,⁃Multiple-target anti-miRNA antisense oligodeoxyribonucleotides; MTg-AMOs; [154].

Such molecules inhibit miRNA biogenesis or interactions of oncogenic miRNA with target mRNA. For example, a synthetic miRNA molecule (AMOs/amiRNA) in the form of an RNA oligonucleotide with a length of 21–23 nucleotides is matched complementarily (as an antisense) to the target miRNA, thereby inhibiting the biogenesis of oncogenic miRNA or preventing its binding to mRNA [155]. AMOs/amiRNAs cause the formation of miRNA duplexes or the degradation of miRNAs. LNA technology, which also leads to the degradation of target miRNAs, uses modified oligonucleotides, in which the ribose ring is “blocked” by a methylene bridge connecting the 2′-O atom and the 4′-C atom. The introduced modification causes stiffening of ribose in the C3′ endo conformation, which significantly improves the in vivo stability of the resulting heteroduplexes [156,157]. The MTg-AMOs strategy simultaneously uses several types of synthetic miRNA molecules to inhibit the expression of different miRNAs concurrently. An in vitro model of stomach cancer used MTg-AMOs, which inhibited the expression of miR-21, miR-106a and miR-221, resulting in inhibition of cancer cell proliferation and migration [158]. miRNA sponges are transcripts containing sites that mimic sequences found in mRNA complementary to the target miRNA. The use of miRNA sponges allows the reduction in the number of free miRNAs (of one or more types) by binding them to a sponge. This leads to an increase in mRNA expression inhibited by “stopped” miRNAs. For example, inhibition of miR-9 (which in the case of breast cancer is characterized by overexpression) through the use of miRNA sponges (mimicking mRNA for cadherin-1) resulted in a reduction in metastasis [158,159]. 

The functional action of miRNA masks is based on inhibition of miRNA interaction with a specific mRNA by competitively binding the introduced artificial miRNA to the end of the 3′ mRNA. Binding of artificial miRNA leads to inhibition of mRNA expression [160]. A limitation of the possibility of the therapeutic use of miRNA molecules is the lack of effective methods for delivering synthetic miRNAs, e.g., to the tumor environment. Chemically synthesized miRNAs can be introduced into cancer cells through various types of transporters, e.g., liposomes and nanocarriers, with the help of transfection reagents or by electroporation [161,162]. In order to develop an effective way of delivering synthetic miRNA variants, the focus should be on ensuring that they are protected from early degradation in the bloodstream, reach the target cells directly and do not trigger an immune response [163,164]. Chemical modification of miRNA oligonucleotides leads to increased stability of such molecules and prevents their degradation by nucleases in the circulation. For example, adding a methyl or methoxyethyl group to the 2′-OH group (in ribose) or replacing the 2′-OH group with a fluorine atom in ribose residues leads to increased stability and affinity of anti-miR binding to the target miRNA, which prolongs the duration of action in organs such as the liver, lungs or kidneys [165]. There is so-called local and systemic delivery of synthetic miRNAs. Local miRNA transport involving the administration of miRNA molecules to the tumor may result in suppression of target genes while maintaining reduced toxicity. This method is more effective compared to the delivery of synthetic miRNAs via the systemic route. However, local miRNA transport is limited to readily available solid cancers such as breast or cervical cancer. Topical application of synthetic miR-145 also produced anticancer effects in mouse models of colon cancer [166]. Attempts to deliver synthetic let-7 (let-7g) using a polymer carrier in a mouse model of lung cancer resulted in a 60–70% reduction in lung tumor [166]. Studies have shown that both routes of delivery (intravenous and directly to the tumor) of synthetic miRNA variants (let-7a) led to a reduction in tumor size in a mouse model of non-small-cell lung cancer [167]. Another example of the methods being tested is the intranasal administration of a lentivirus vector expressing let-7a in a model of human non-small-cell lung cancer, resulting in inhibition of the growth of KRAS-dependent lung tumors. 

Potential therapeutic use may have RNA aptamers [168,169]. Aptamers are short, single-stranded nucleic acid molecules that adopt a strictly defined tertiary structure, which is responsible for the selectivity of their interaction with specific ligands, e.g., with nucleic acids, peptides, proteins or low-molecular compounds. Tumor suppressor miRNAs have gained importance in anticancer therapies. The generation of the aptamer-miRNA chimera to direct the delivery of let-7g miRNA to the cancer cell has been described [170]. GL21. T aptamer was used, which blocks the activity of the oncogenic tyrosine receptor Axl kinase and demonstrates the specificity of GL21. Targeted T-let aptamer-miRNA chimera by increased let-7g accumulation in Axl-expressing A549 cells, but not in cells lacking AXL MCF-7. After systemic administration of aptamer-miRNA chimera in immune-deficient mice with an A549 NSCLC-derived tumor, there was downward adjustment of the target 7g HMGA2, as well as inhibition of tumor growth [170].

A strategy was also developed using the same aptamer, although this time it was converted to miR-212 to deliver it to A549 NSCLC cells (non-small-cell lung cancer). Research demonstrated that GL21 T-miR-212 increases caspase activation in NSCLC and enhances the effect of such a strategy in combination with TRAIL therapy as an adjuvant [171]. 

Another study used a transferrin receptor aptamer (TRA) to deliver miR-126 to endothelial cells and breast cancer. A chimera consisting of a TRA coupled by a sticky bridge to pre-miR-126 was generated. Pre-miR-126 was successfully delivered to human endothelial cells and MCF7 breast cancer cells in vitro. The effect of VCAM-1 (vascular cell adhesion molecule 1) was inhibited, reducing the proliferation of MCF7 cancer cells, as well as reducing the number of paracrine endothelial cells in the same breast cancer model in vitro. The effect achieved by this is comparable to the effect obtained by transfection of miR-126-3p via liposomes [172].

A new aptamers-based strategy has been investigated that alleviates hematopoietic toxicity induced by chemotherapeutic drugs, such as carboplatin and 50 fluorouracil (5-FU), in hematopoietic stem/progenitor cells (HSPC). A chimera consisting of an aptamer c-Kit coupled with miR-26a mimic was generated to restore Bak1 (Bcl-2 antagonist/killer) to HSPC. The importance of restoring BAK1 in chemotherapy-induced myeloablation as well as after bone marrow transplantation using a chimera mimicking c-Kit-miR26a has been demonstrated. In vivo experiments in MICE NOD scid gamma NSG have shown that systemic administration of aptamer-microRNA chimera protects individuals from carboplatin and 5-FU while inhibiting tumor growth in the MDA-MB-231 human tumor model [173]. Figure 4 shows RNA aptamers used to treat cancer.

All the strategies presented above give us hope for a more effective fight against cancer, but it is worth paying attention to the doubts associated with them. First, the study of one selected miRNA molecule is in most cases insufficient, because the changes occurring in cancer cells are dependent on the expression of various pleiotropic miRNAs. Secondly, the in vivo delivery of artificial miRNAs is a challenge, and no single, effective method of transporting them to the target tissue (specific route of delivery) has yet been proposed [174]. Thirdly, the use of synthetic miRNA carriers raises doubts, due to the possibility of their degradation in the blood and potential toxicity [175,176,177,178] and the durability of the effects of therapy.

## 6. Summary

Since the discovery of miRNA molecules, interest in them has been increasing. Their role is still not fully understood, but there is no doubt that they have a huge impact on the cell and its physiology. Dysregulation of the level of expression of miRNAs present in tissues, as well as in the blood, are found in almost all diseases, which shows their significant participation in pathological processes. The presence and stability of circulating miRNAs in easy-to-collect blood and sputum make the use of these molecules as noninvasive biomarkers very promising. Already, the results so far suggest that miRNA molecules can contribute to the development of screening for early detection of cancer, and by determining the level of risk, which is a huge concern in the era of personalized medicine, miRNA can significantly improve the quality of treatment for cancer patients.

## Figures and Tables

**Figure 1 ijms-23-02805-f001:**
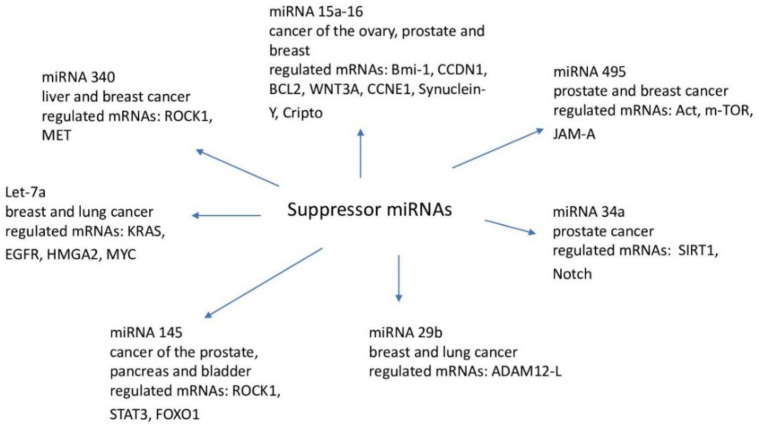
Suppressor miRNA.

**Figure 2 ijms-23-02805-f002:**
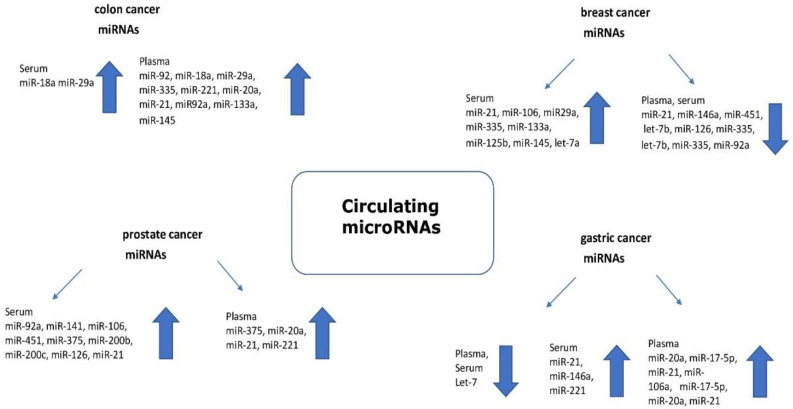
Circulating microRNAs as potential cancer biomarkers (the arrow indicates the expression level).

**Figure 3 ijms-23-02805-f003:**
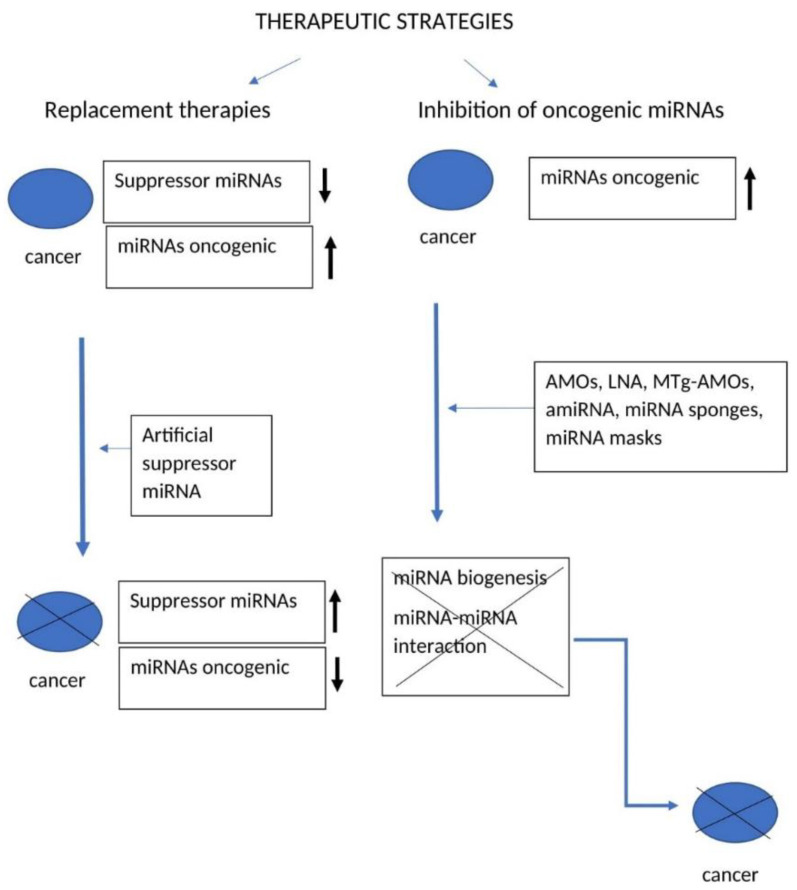
Schematic compilation of anticancer strategies using miRNA.

**Figure 4 ijms-23-02805-f004:**
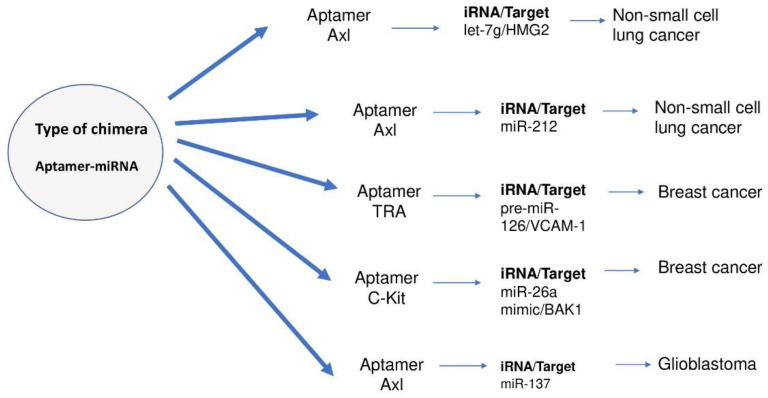
RNA aptamers in the treatment of cancer [170,171,172,173].

**Table 1 ijms-23-02805-t001:** Compilation of circulating miRNAs taking into account the place of occurrence in the body and the diagnostic and prognostic potential.

Type of Cancer	Diagnostic Marker	Prognostic Marker
breast cancer	let-7c [20], miR-10 [21] miR-16 [22] miR-18a [23], miR-106b [24], miR-21 [25], miR29a [26], miR-34a [27], miR34b/c [28], miR-125b [29], miR-155 [30]	miR-18a [31,32], miR-106b [33], miR-21 [34], miR-34a [35], miR-125a [36], miR-125b [37], miR-155 [38]
lung cancer	let-7a [39], let-7c [40], let-7f [41], miR-10 [42], miR-19b [43], miR-21 [44], miR29c [45], miR-30a [46], miR-30c [47], miR-34c [48], miR-155 [49], miR-200c [50]	let-7i [51], miR-10 [52], miR-19b [53], miR-21 [54], miR-30d [55], miR1-25b, miR-200c, miR-210 miR-375, miR-429 [56]
liver cancer	miR-16 [57], miR-21 [58]	miR-21 [58]
gastric cancer	let-7a [59], miR-18a [60], miR-106a [60], miR-106b [60], miR-21 [60]	miR106a [60], miR-21 [60]
colorectal cancer	let-7a [61], let-7f [62], miR-18a [63], miR-19a [64], miR-20a [65], miR-19, miR-92 [66], miR-92a [67], miR-106a [68], miR-21 [69], miR-221 [62]	miR17 [70], miR-21 [71], miR-29b [71], miR-29c [71]
pancreatic cancer	miR-205, miR-21, miR-642b, miR-885-5p, miR-22 miR-145, miR-150, miR-223, miR-636, miR-26b, miR-34a, miR-122, miR-126, miR-145, miR-150, miR-155, miR-376a, miR-301, miR-223, miR-505, miR-636, miR-885.5p [72,73]	miR-130b, miR-21, miR-105, miR-196a-2, miR-221, miR-203, miR-210, miR-222, miR-452, miR-105, miR-127, miR-187, miR-518a-2, miR-30a-3p [74,75]
prostate cancer	miR-30c, miR-622, miR-1285, miR-10b, miR-373, let-7c, -7e miR-141, -375, miR-26a, -195 [76,77]	miR-141, miR-375, miR-20a, miR-21, miR-141, miR-145, miR-125b, miR-224, miR-23b, miR-222, miR-221 [78,79]
ovarian cancer	miR-200 family, let-7 family, miR-21, miR-29a, miR-92, miR-93, miR-126, miR-127, miR-132, miR-155, miR-214, miR-182, miR-205, miR-144, miR-145, miR-222, miR-302 [80,81,82]	miR-410, -645, miR-199a, miR-200 family, miR-140, miR-141, -429 [83]
skin cancer	let-7a, b, miR-148, miR-155, miR-182, miR-203, miR-205, miR-200c, miR-211, miR-214, miR-221, miR-222, miR-150, miR-342-3p, miR-455-3p, miR-145, miR-155, miR-497 [84]	miR-221, miR-199a-5p, miR-33a, miR-424, miR-16, miR-125b, miR-200c, miR-205, miR-142-5p, miR-150-5p, miR-342-3p, miR-155-5p, miR-146b-5p [84]
kidney cancer	miR-141, miR-224, miR-21, miRNA-32, miR-34a, miRNA-203, miR-378, miR-210, miR-20b-5p, miR-30a-5p, miR-196a-5p, miR-224-5p, miR-34b-3p, miR-182-5p, miR-210, miR-508-3p, miR-885-5p, miR-210, miR-378, miR-451, miR-21, miR-106a, miR-200a, miR-193a-3p, miR-362, miR-572, miR-28-5p, miR-378 [85]	miR-122-5p, miR-206, miR-21-5p, miR-210-3p, miR-150, miR-210, miR-221, miR-1233, miR-7, miR-221, miR-222, miR-221, miR-224, miRNA-15a, miR-17-5p–miR-25-3p, miR-let-7i-5p, miR-26a-1-3p, miR-615-3p [85]
thyroid cancer	miR-146b, miR-221, miR-222, miR-15a, miR-155 [86]	miR-146b, miR-221, miR-222 [86]
non-small-cell lung cancer (NSCLC)	let-7c, miR-138, miR-145, miR-183, miR-29 family, miR-34a, miR-34c-3p, miR-101-3p, miR-129, miR-200b, miR-212, miR-218, miR-449a, miR-45165, miR-21/155, miR-25, miR-31, miR-221/222, miR-224, miR-191, miR-494, miR-19a, miR-34697 [87]	miR-1290, miR-1246, miR-150, miR-21-5p [88]
B-cell lymphoma	miR-17/92, miR-106a-363, miR-200c-3p, miR-638, miR-518a-3p, miR-17-5p, miR-217-5p, miR-634, miR-26b-5p, miR-330-3p, miR-106a-5p, miR-210-3p, miR-612, miR-188-5p, miR-302c-3p, miR-433-3p, miR-584-5p, miR-200a-3p, miR-135a-5p, miR-375-3p, miR-138-5p, miR-517 isomiRs, miR-330-3p, miR-106a-5p, miR-210-3p, miR-301 isomiRs, miR-338-5p [89]	miR-144-3p, miR-431-5p, miR-376b-3p [89]

**Table 2 ijms-23-02805-t002:** Tumor-associated miRNAs.

miRNA	Tumor Type	Target Genes
let-7	NSCLC	RAS
miR-21	colorectal cancer, cancer of the stomach, lung cancer	MYCN, ATM, FXR, EGR2, MXD1, PIAS3, SOCS6, HIF-1a
miR-17-92	breast cancer	AIB1 (miR-17-5p), E2F1 (miR-17-5p, miR-20a), TGFBR2 (miR-20a), Tsp1 and CTGF
miR-106a	colorectal cancer, pancreatic cancer, prostate cancer	Rb1
miR-221, miR-222, miR-146b	thyroid, papillary cancer	KIT
miR-182	lung cancer	Rsu1, Mtss1, Pai1, Timp1
miR-155	colorectal cancer, lung cancer, pancreatic cancer	RAD51, VHL, SOCS1
miR-372, miR-373	testis, germ cell tumors	LATS2
miR-221/222	stomach cancer, prostate cancer	p27, PTEN

## Data Availability

Data sharing is not applicable to this article.

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
