# Peer review of "miRNAs in Cancer (Review of Literature)"

_ijms, 2022, doi:10.3390/ijms23052805_

Round 1
Reviewer 1 Report
The role of miRNAs in oncogenesis has received significant attention in the last decade and there are many systematic reviews on the role of miRNAs as “tumor suppressor miRNAs” or “onco-miRs”.
In the review "miRNA in Cancer", the authors describe the current knowledge on the role of the miRNAs in various cancers but I would have suggestions to make it more interesting and innovative:
- Shorten the chapter 2.
- Add tables or diagrams to the various chapters to make the text simpler and more impressive.
- There has been increasing interest and investigation into miRNAs as potential prognostic biomarkers for cancer over the last decade. Add to the review the literature of miRNA signatures and their value in predicting the malignant transformation in many cancers.
- Despite the therapeutic potential of miR-based molecules, the development of efficient delivery strategies is a key aspect for their development. Add to the chapter 5 data on the role of aptamer-miRNA chimeras as potential therapeutic approach.
- Update the text and the bibliography. There are only 3 references out of 101 of the year 2021
Author Response
Thank you for your review.
I would like to kindly ask you to reconsider the publication of our revised paper:
" miRNAs in Cancer (Review of Literature)”.
I hereby provide responses to the reviewers and list the changes that have been made in the revised version of our paper.
Reviewer 1
The role of miRNAs in oncogenesis has received significant attention in the last decade and there are many systematic reviews on the role of miRNAs as “tumor suppressor miRNAs” or “onco-miRs”. In the review "miRNA in Cancer", the authors describe the current knowledge on the role of the miRNAs in various cancers but I would have suggestions to make it more interesting and innovative:
1. Shorten the chapter 2.
Answer. Chapter 2 has been completely rebuilt and shortened
2. Add tables or diagrams to the various chapters to make the text simpler and more impressive.
Answer. Tables and diagrams have been added. They are present in every section.
3. There has been increasing interest and investigation into miRNAs as potential prognostic biomarkers for cancer over the last decade. Add to the review the literature of miRNA signatures and their value in predicting the malignant transformation in many cancers.
Answer. miRNAs as potential biomarkers are described in Chapters 2 and 3, and Tables 1 and Figure 1 in this topic have been added.
4. Despite the therapeutic potential of miR-based molecules, the development of efficient delivery strategies is a key aspect for their development. Add to the chapter 5 data on the role of aptamer-miRNA chimeras as potential therapeutic approach.
Answer. miRNA aptamers are described in section 5
5. Update the text and the bibliography. There are only 3 references out of 101 of the year 2021
Answer. 53 literature items from 2020-2021 have been placed
Reviewer 2 Report
Smolarz et al. present a review article on microRNAs in cancer. Although the authors describe a lot of information on miRNAs, the contents of the first part of the manuscript on the basic knowledge of miRNAs- biogenesis of miRNAs, mechanism of action, and circulating miRNAs (sections 1-3) are not novel. In fact, the cited articles in that part are not the recent ones except for one review article (no.10). There exists already an excellent review article in Frontiers in Endocrinology, “Overview of MicroRNA Biogenesis, Mechanisms of Action and Circulation” published by J. O’brien et al. in 2018. In addition, the authors have already published another review article, “The role of microRNA in pancreatic cancer”, in Biomedicines (2021, 9: 1322). The content of that review overlaps some of the materials described in the present manuscript, and the quality of that review seems to be much better.
Anyway, I strongly recommend that this introductory part of the manuscript must be presented more concisely, and the arguments of the review must be focused on the miRNAs in cancer, as is expected from the title.
I have a little concern how much up-dated information is included in the sections 4 and 5, related to the role of miRNAs in oncogenesis and the application in cancer diagnosis and therapy. I find 9 out of 12 citated papers recently published (2019-2021) in the reference section are review articles (references No: 39, 42, 45, 47, 63, 64, 73, 83and 89). A review of recently published review articles certainly do not help readers to get novel information in the field.
Minor comments
- on line 36, antisensically ?
- line 47, conserved ?
- line 323, mimic mimics
Author Response
Thank you for your review.
I would like to kindly ask you to reconsider the publication of our revised paper:
" miRNAs in Cancer (Review of Literature)”.
I hereby provide responses to the reviewers and list the changes that have been made in the revised version of our paper.
Answer. We agree with the review. The second chapter was completely shortened and rebuilt. Information about miRNA biogenesis has been removed because, as the reviewer noted, it was included in our earlier work - The role of microRNA in pancreatic cancer. We tried to make the chapter concise and deal with the topic contained in the title. New literature from 2020-2021 has also been published, mainly these are original works. We have introduced additional tables and figures that make the text simpler and more impressive. Error in line 323 has been corrected.
Round 2
Reviewer 1 Report
I really appreciated the changes made to the text. I believe that in this new version the manuscript can be published
Author Response
Reviewer 1
Thank you very much for your positive review
Reviewer 2 Report
The review article summarizes biological function of miRNAs in normal tissues and cancer. Particularly, the authors focused on the circulating miRNA as a biomarker and their role in oncogenesis. In addition, the authors describe a potential use of synthetic miRNA derivatives and RNA aptamers, exogenously introduced, to regulate levels of endogenous miRNAs in the field of cancer therapy. Respect to the previously presented manuscript, general description on the miRNA biogenesis is now more concise and original articles are inserted in the” References” instead of citing other review articles. Thus, evidently I find some progress in the revised manuscript. However, there are some concerns left in the way of presentation of the manuscript. In particular,
- I think it much more logical to put the Section 4 (miRNAs and their role in Oncogenesis) before the Section 3 (Circulating miRNA), as the story of the circulating miRNA is directly related to the argument on the diagnosis, presented in the Section 5 (microRNA in cancer diagnosis and therapy).
- In the Table 1, the authors present circulating miRNAs as candidates for diagnostic and prognostic markers in breast, lung, liver, gastric and colorectal cancers. In the previous review article published by the same authors (The role of microRNA in Pancreatic Cancer. Biomedicines 2021, 9: 1322), there is a list of miRNAs present in various types of cancer (Table2). However, I find only a few miRNAs that are overlapped with the ones listed in the Table 1 of the present manuscript. Is it simply because some of the miRNAs are not found in circulation, and/or there is no published data ? Why the circulating miRNAs identified in pancreatic cancer, described in the previous review article, are not included in the Table1 ? The issues must be explained appropriately.
- Table 1. could be improved by addition of other circulating miRNAs found in pancreatic cancer, B-cell lymphoma, NSCLC, ovarian cancer, prostate cancer, skin cancer, kidney cancer, thyroid cancer, and so on.
- Some of the expressions in English are not precise enough to let readers to understand the concept. Some of them are listed in minor points. Overall editing of English language is absolutely necessary.
Minor points:
- line 15, “and finally cancer” must be deleted.
- “by the ability to encode information about proteins” on line 31: “characterized as non-coding RNAs”
- line 49, “age of pregnancy”, could be “stage of pregnancy” ?
- lines 78-79, “unchanged disease”? “identification of miRNAs specifically expressed in tumor tissues, or miRNAs whose expression level is different from that present in normal tissues”.
- lines 155-158, classification of the miRNAs in “qualitative” or “quantitative” is not appropriate.
- line 259, “miRNA therapies are recently considered as a cancer treatment strategies” ?
Author Response
Reviewer 2
Thank you very much for your review
As suggested we have placed section 4 before section 3.
In Table 1, we have placed miRNAs as diagnostic and prognostic markers based on the latest literature data. Therefore, the miRNAs found in Table 2 in the article The role of microRNA in Pancreatic Cancer. Biomedicines 2021, 9: 1322 may not have been included in this table because, according to current literature data, they are not included in any group of markers. We tried to make Table 1 based on literature data from 2020-2021. We also expanded the table to include miRNAs in other cancers It seems to us that it is extensive and contains the most important data for today.We tried to make the language of the article easy and precise. We would like to mention that the second appointed reviewer had no comments in this area.
Minor revision have been corrected
Round 3
Reviewer 2 Report
Style of the manuscript has been revised appropriately and its quality has been improved. However, there are still some inappropriate expression in English. In particular,
- lines 178-179, “as well as in the treatment of cancer patient” is repeated.
- line 205, Vakadu et al.(110) In their----.
- line 223, from which type of cells they come from
- line 235, “all body fluids”, isn’t it better to say “various body fluid”
- lines 296-297, “consists in---“, Inhibition of oncogenic miRNAs can be obtained by the introduction of synthetic DNA or RNA molecules that mimic onco-suppressive effects in cancer cells”
- line 315, “durability” could be substituted by “in vivo stability” ?
- lines 327-328, “inhibition of miRNA interaction with---“
- line 367, “immune deficient mice carrying A549 NSCLC derived tumor” ?
- line 405, “Dysregulation” ?
Anyway, it is strongly recommended that the manuscript will be edited by a specialized person in scientific English writing.
Author Response
Thank you for your review.
I would like to kindly ask you to reconsider the publication of our revised paper:
" miRNAs in Cancer (Review of Literature)”.
I hereby provide responses to the reviewers and list the changes that have been made in the revised version of our paper.
Reviewer 2
Style of the manuscript has been revised appropriately and its quality has been improved. However, there are still some inappropriate expression in English. In particular,
- lines 178-179, “as well as in the treatment of cancer patient” is repeated.
Has been corrected
- line 205, Vakadu et al.(110) In their----.
Has been corrected
- line 223, from which type of cells they come from
Has been corrected
- line 235, “all body fluids”, isn’t it better to say “various body fluid”
Has been corrected
- lines 296-297, “consists in---“, Inhibition of oncogenic miRNAs can be obtained by the introduction of synthetic DNA or RNA molecules that mimic onco-suppressive effects in cancer cells”
Has been corrected
- line 315, “durability” could be substituted by “in vivo stability” ?
Has been corrected
- lines 327-328, “inhibition of miRNA interaction with---“
Has been corrected
- line 367, “immune deficient mice carrying A549 NSCLC derived tumor” ?
Has been corrected
- line 405, “Dysregulation” ?
Has been corrected